# A Study on the 3D Reconstruction Strategy of a Sheep Body Based on a Kinect v2 Depth Camera Array

**DOI:** 10.3390/ani14172457

**Published:** 2024-08-23

**Authors:** Jinxin Liang, Zhiyu Yuan, Xinhui Luo, Geng Chen, Chunxin Wang

**Affiliations:** 1Institute of Animal Science and Veterinary Medicine, Jilin Academy of Agricultural Sciences, Gongzhuling 136100, China; liangjinxin0124@163.com (J.L.);; 2College of Animal Science and Technology, Jilin Agricultural University, Changchun 130118, China

**Keywords:** Kinect v2, computer vision, 3D reconstruction, sheep body, point cloud processing

## Abstract

**Simple Summary:**

Traditional methods for measuring animal body size primarily rely on manual data collection, which is not only costly and inefficient but also prone to significant data errors and can cause stress to the animals. To address these issues, this study introduces a 3D point cloud reconstruction method for sheep body measurement, achieving the synchronous collection and reconstruction of sheep body point cloud data. For the first time, a conditional voxel filtering box is proposed for downsampling, which effectively reduces the number of point clouds. A rotation normalization algorithm is used to correct the three views. Finally, a comparison is made between manual measurement speed, system measurement speed, and normalized system measurement speed. The results show that the normalized processing group has the most ideal time, with an average of 0.74 min to measure six indicators for one sheep. Lastly, the accuracy of the 3D point cloud reconstruction of the sheep body is verified, with a reconstruction accuracy error of 0.79% for body length as an example, indicating that the non-contact body measurement method based on 3D point cloud reconstruction is feasible and can provide an important reference for the breeding of superior breeds.

**Abstract:**

Non-contact measurement based on the 3D reconstruction of sheep bodies can alleviate the stress response in sheep during manual measurement of body dimensions. However, data collection is easily affected by environmental factors and noise, which is not conducive to practical production needs. To address this issue, this study proposes a non-contact data acquisition system and a 3D point cloud reconstruction method for sheep bodies. The collected sheep body data can provide reference data for sheep breeding and fattening. The acquisition system consists of a Kinect v2 depth camera group, a sheep passage, and a restraining pen, synchronously collecting data from three perspectives. The 3D point cloud reconstruction method for sheep bodies is implemented based on C++ language and the Point Cloud Library (PCL). It processes noise through pass-through filtering, statistical filtering, and random sample consensus (RANSAC). A conditional voxel filtering box is proposed to downsample and simplify the point cloud data. Combined with the RANSAC and Iterative Closest Point (ICP) algorithms, coarse and fine registration are performed to improve registration accuracy and robustness, achieving 3D reconstruction of sheep bodies. In the base, 135 sets of point cloud data were collected from 20 sheep. After 3D reconstruction, the reconstruction error of body length compared to the actual values was 0.79%, indicating that this method can provide reliable reference data for 3D point cloud reconstruction research of sheep bodies.

## 1. Introduction

With the rapid development of computer vision technology, the application of 3D reconstruction technology in the livestock industry is gradually increasing [1]. Depth camera-based 3D reconstruction technology has gradually become a research hotspot, providing new solutions for animal measurement [2]. Traditional methods of animal body measurement typically involve manual measurement, which is susceptible to subjective factors. These methods are time consuming, labor intensive, prone to errors, and can easily cause stress to the animals, thereby affecting their welfare [3,4]. Considering these factors, researchers have proposed various improved methods.

To reduce the stress response of animals during the body measurement process, Ruchay et al. designed an automated computer vision system capable of capturing accurate 3D models of live cattle. This system excels in the accuracy and efficiency of body measurements and can complete measurement tasks without interfering with the normal activities of the cattle, greatly enhancing the convenience of measurement and animal welfare [5].

Machine measurement can effectively reduce human operational errors. Shuai et al. used an RGB-D camera to perform 3D surface reconstruction and body measurement using multi-view images. By obtaining depth and color information through the RGB-D camera, the research team constructed a 3D model of the livestock and calculated the body measurement data through algorithms. This system demonstrated high accuracy in trials, effectively reducing human operational errors and animal stress responses and significantly improving the efficiency and accuracy of measurements [6].

The accuracy of machine body measurements is an important aspect we need to focus on. Guo et al. studied an interactive 3D point cloud analysis software (LSSA_CAU) for measuring the body dimensions of livestock, such as cattle or pigs. The application of LSSA_CAU software not only improved measurement efficiency but also processed large amounts of data in a short time, suitable for the daily management of large-scale farms. The results showed that this software performed excellently in practical applications, significantly enhancing the automation and accuracy of body measurements.The updated version of LSSA_CAU can be downloaded for free from the following link: https://github.com/LiveStockShapeAnalysis (accessed on 1 July 2024) [7].

With the development of computer technology, 3D reconstruction methods are continuously innovating. Jia et al. studied a template matching-based algorithm for cow body extraction and segmentation based on depth images. This algorithm automatically segments the depth images of cow bodies and extracts key points from the segmented regions [8]. The results showed that this algorithm had significant advantages in the segmentation accuracy of various parts of the cow body, providing reliable data support for subsequent body measurements and weight estimation.

Traditional animal body measurement methods primarily rely on manual data collection, which is not only costly and inefficient but also prone to significant data errors and can cause stress to the animals [9,10]. To address these issues, this study proposes a 3D point cloud reconstruction method for sheep bodies, achieving synchronous acquisition and reconstruction of sheep body point cloud data. The reconstruction accuracy meets the relevant body measurement standards, providing a theoretical basis for the measurement of sheep body phenotype data [11].

## 2. Material and Methods

This experiment designed a three-view depth data acquisition system specifically for sheep to achieve a 3D point cloud reconstruction of sheep. Tölgyessy uses a set of depth cameras and specially designed stanchions in conjunction with traditional sheep passages to simultaneously capture the depth maps of sheep from multiple perspectives [12]. The experiment was conducted at the sheep breeding site of the Jilin Academy of Agricultural Sciences. Twenty meat sheep, aged 8–12 months and weighing between 35 and 45 kg, were selected and individually housed for the experiment.

### 2.1. Design of the Restraining System

The system mainly consists of a Kinect v2 depth camera set, a sheep passage, a restraining pen, and a background cloth. To accommodate the angles for image acquisition by the depth cameras, the restraining pen is designed in a trapezoidal shape. The background cloth is made of scratch-resistant and durable PVC material, with a color significantly different from that of the sheep, specifically green. A total of three Kinect v2 depth cameras are deployed: one installed at the top of the trapezoidal restraining pen, one at the shortest parallel side of the trapezoid, and one at the side of the trapezoid. These cameras are used to capture depth images of the sheep from top, side, and front views, respectively. The maximum horizontal shooting angle of the Kinect v2 camera is 70°, and the optimal shooting distance for the front view is determined to be 0.9 m. Therefore, the trapezoidal restraining pen is designed with an acute angle of 53°. The system structure and physical diagram are shown in Figure 1a–c.

### 2.2. System Parameters

To achieve the collection of multi-view depth data of sheep in their natural state and reduce the potential stress caused by manual measurement of body dimensions and weight, a combination of a passage and a restraining pen was used for data collection. This setup minimizes space while allowing adult rams and ewes of different sizes to pass through smoothly. After testing at the experimental sheep farm, the final system parameters were determined, as shown in Table 1.

The above parameters ensure that adult rams and ewes can pass through the passage and restraining pen smoothly without turning around or backing up. Additionally, they ensure that when the sheep enter the restraining pen, clear and usable depth images from three angles can be captured.

## 3. Sheep Body Point Cloud Data Acquisition

### Data Collection Process

This study proposes a non-contact data acquisition system and a method for reconstructing 3D point clouds of sheep bodies. Three Kinect v2 depth cameras are used simultaneously to capture depth images from three different angles for the 3D point cloud reconstruction of sheep bodies. The block diagram of the proposed acquisition system is shown in Figure 2. The simultaneous operation of multiple depth cameras is achieved using the C++ programming language and the API provided by the Kinect SDK to write a synchronized acquisition program [13].

Before the experiment, 20 sheep were manually measured for body parameters, weighed, and marked. Ear tags and related data were compared for evaluation and analysis of the reconstruction effect. The acquisition equipment was installed in a separate sheep pen, and sheep were slowly guided into a restraint channel manually. When a sheep entered a specific restraint pen, the entrance was closed, and the three depth cameras simultaneously collected depth data. During the experiment, sheep were driven into the restraint channel to ensure they could move naturally without being constrained, while also preventing them from walking side by side, turning around, or returning. Each sheep passed through the restraint pen several times in a natural state, and a total of 157 sets of depth data were collected.

## 4. 3D Reconstruction of the Sheep Body

Convert the depth maps collected by the Kinect v2 camera array into point cloud data. First, use the Zhang Zhengyou calibration method by Xiong et al. to transform the three views of the depth camera into the same world coordinate system. After applying rotation and translation transformations, obtain usable depth maps from the three views [14].

### 4.1. Point Cloud Preprocessing

Due to the complex environment in which the depth maps are collected and the presence of numerous irrelevant noise points, the subsequent registration operations are significantly affected [15]. It is necessary to perform further accurate and effective operations on the obtained point cloud data to remove a large amount of irrelevant noise points [16]. This involves preprocessing the point cloud data, which mainly includes point cloud denoising and point cloud repair [17].

#### 4.1.1. Point Cloud Denoising

For the useless noise points collected, such as limit devices, ground point clouds, and outliers, it is necessary to perform denoising on the redundant noise points [18]. This study uses passthrough filter, statistical outlier removal, and random sample consensus to handle the noise points. Additionally, conditional removal and a voxel grid gilter are used in combination to downsample and simplify the point cloud data [19].

(1)Passthrough Filter: This filter is based on the defined range of points along the x, y, and z axes. It removes unnecessary points to retain the region of interest, thereby obtaining the main body of the sheep within the point cloud channel.(2)Statistical Outlier Removal: To eliminate the influence of outlier noise points, statistical filtering is used to remove abnormal points around the sheep, i.e., the elimination of discrete isolated points in the point cloud. This filter is based on the Gaussian distribution characteristics of the points. First, for each point in the point cloud, the k-nearest neighbors are calculated, and the Euclidean distances from these neighbors to the point are computed. The average of these distances is taken as the characteristic distance of the point. Next, a threshold is set based on statistical principles to remove abnormal points: all characteristic distances are statistically analyzed using the mean and standard deviation. A threshold R, which is the mean plus the standard deviation, is set. Points with distances exceeding this threshold are considered abnormal and are removed. In this study, the threshold R is set to 0.6. The effects before and after filtering are shown in Figure 3a,b.(3)Random Sample Consensus (RANSAC): This algorithm is a ground segmentation algorithm that uses random sampling and iterative optimization of subsets to estimate model parameters. Three data samples are randomly selected from the point cloud data to form a plane. The distance d from any data point in the point cloud to this plane is calculated. If d is less than the distance threshold, the point is considered an inlier, i.e., it lies within the plane. Multiple iterations are performed to remove all outliers from the point cloud data.

To evaluate this algorithm, this study conducts ground point cloud removal experiments with a maximum iteration count k of 550 and distance thresholds α set to 0.015, 0.02, and 0.03. The experimental results are shown in Figure 3c–e. In Figure 3c, denoising is incomplete, with some point clouds remaining; Figure 3d shows the best denoising effect, with no large areas of ground point clouds; and Figure 3e shows partial local point cloud loss in the sheep’s limbs due to over-denoising. From this analysis, it can be concluded that setting the distance threshold α to 0.02 effectively removes ground point cloud noise points.

(4)After the above filtering processes, the problem of large point cloud data volume still leads to slow computation by the computer. Based on this, this study proposes a conditional voxel filter box, which is the first to combine the advantages of conditional filtering and voxel filtering for animal downsampling to uniformly thin out the point cloud data.

Conditions are set based on the intensity of the point cloud in conditional filtering. Areas with darker data colors have denser point clouds, making them the targeted working areas for conditional filtering. Voxel filtering will focus on downsampling these areas. The principle is to create a three-dimensional grid based on the input point cloud data, as shown in Figure 3. The size and number of grids are related to the required number of point clouds; the more grids there are, the more filtered point clouds, and the more evident the feature retention. Then, the centroid of the point cloud within each sub-grid space is calculated to replace the entire point cloud within the grid, thereby achieving the goal of downsampling the target point cloud while preserving its features as much as possible. The process is as follows.

**Figure 3 animals-14-02457-f003:**
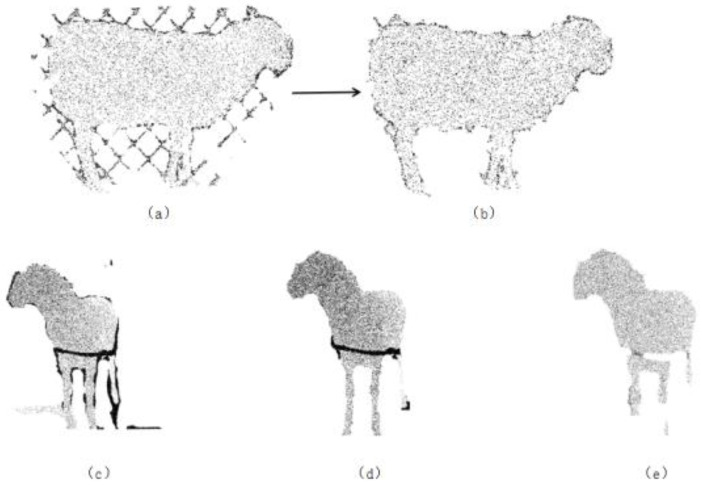
The effect of using statistical filtering and random sample consensus algorithms, where (**a**,**b**) are comparisons before and after statistical filtering; (**c**–**e**) are comparisons with random sample consensus thresholds α of 0.015, 0.02, and 0.03, respectively.

First, calculate the minimum bounding rectangle suitable for sampling the point cloud, as shown in Figure 4.

Second, this study specifies the voxel grid size, as shown in Figure 4b. According to the formula in the figure, a total of N voxel grids is obtained [20].
(1)$$\left\{\begin{array}{l} L=l_{x}/\lambda \\ W=l_{y}/\lambda \\ H=l_{z}/\lambda \end{array}\right\}\Rightarrow N=L•W•N$$

In Equation (1), using the x, y, and z coordinates as the baseline, L is the number of grids along the x axis; W is the number of grids along the y axis; and H is the number of grids along the z axis.

Finally, the centroid of the points within each cell is calculated using the following formula and replaces the other point clouds within its voxel grid. The result is shown in Figure 4c.
(2)$$c_{i}=\frac{1}{n}\sum_{i}^{n}P_{i}$$

In Equation (2), n is the number of point clouds within the voxel grid, and P_i_ is the point within the grid.

As shown in Figure 5, with the voxel grid size set to 0.5, the use of conditional voxel filtering effectively retains the feature regions and reduces the point cloud data volume. This downsampling of the point cloud effectively alleviates the computational workload in subsequent processing.

The use of passthrough filtering, statistical filtering, and random sample consensus can effectively remove useless points such as occlusions, outliers, and ground noise points. The combination of conditional and voxel filtering not only retains the main features but also eliminates redundant point cloud data. This provides substantial data support for subsequent steps, such as 3D reconstruction and feature extraction.

#### 4.1.2. Point Cloud Repair

In this study, due to occlusions from the limiting bars and other uncertain factors, there are missing parts in the frontal view data of the sheep’s body after point cloud denoising, which significantly affects subsequent 3D reconstruction. Current research on point cloud repair techniques mainly focuses on plant root systems, terrain restoration, and virtual scene construction. Common point cloud repair methods include interpolation, surface reconstruction, deep learning, multi-view fusion, and global optimization [21,22].

Compared to other existing point cloud repair methods, the cubic spline interpolation method in interpolation can smoothly reconstruct the original point cloud data of the surface while ensuring that the reconstruction patches in the filled areas are consistent with the original surface’s smoothness [23]. All samples at the edges of the missing point cloud regions can be completed during the process. The basic idea is to perform piecewise interpolation using multiple cubic polynomial functions [24]. The general form is as follows:

For each interval $\left(\left[x_{-}i,x_{-}\left\{i+1\right\}\right]\right)$, we have the following:(3)$$S_{i}(x)=a_{i}+b_{i}(x-x_{i})+c_{i}{\left(x-x_{i}\right)}^{2}+d_{i}{\left(x-x_{i}\right)}^{3}$$

In Equation (3), $\left(S_{-}i\left(x\right)\right)$ is the cubic polynomial for the (*i*)-th interval. *a_i_*, *b_i_*, *c_i_* and *d_i_* are the coefficients to be determined, which are solved through the continuity of boundary conditions and the smoothness and derivative continuity conditions. The repair effects are shown in Figure 6, where Figure 6a shows the repair effect after removing occlusions from the limiting bars using the interpolation method and Figure 6b shows the repair effect on the details of the sheep’s hind legs using the interpolation method. Compared to other repair methods, this method yields better results.

### 4.2. 3D Point Cloud Reconstruction from Three Views

Typically, point cloud reconstruction methods achieve registration by obtaining a rotation matrix through multi-camera calibration or by iteratively optimizing the overlapping regions of point clouds to find the best matrix for multi-viewpoint cloud registration [25]. However, these methods often result in low accuracy or relatively complex operations. This study combines feature-based RANSAC for coarse registration followed by ICP for fine registration to improve registration accuracy and robustness [26,27].

#### 4.2.1. RANSAC Algorithm for Coarse Registration

This method estimates the sample data as a mathematical model and divides the data into inliers and outliers based on set threshold parameters. By iteratively removing outliers, the registration accuracy is improved. The main calculation principles are as follows:(1)Randomly select non-collinear point pairs: Randomly select three non-collinear corresponding point pairs (F1, F2, F3) from the point set after removing mismatched point pairs.(2)Solve the transformation matrix: Use the SVD transformation matrix solving method to calculate the transformation matrix H_C_ for F1, F2, and F3.(3)Calculate the mean squared error: Calculate the mean squared error value K between the source point cloud and the target point cloud.(4)Compare errors and iterate: Compare the mean squared error value K with the set error threshold V. If K is greater than K, return to step (1) and continue iterating until the error approaches the threshold or the set number of iterations t is reached.(5)Complete coarse registration: Use the final transformation parameter matrix H to perform coarse registration of the source point cloud with the target point cloud.

#### 4.2.2. ICP Algorithm for Fine Registration

The traditional ICP algorithm iteratively corrects the rigid body transformation matrix of the source and target point clouds to minimize the distance between all point sets. Due to the need for multiple repeated iterations in nearest neighbor search, the algorithm is time consuming.

(1)Assume the input point cloud data are P and Q, with an overlapping part θ. The position of any point in P and Q is PS and Q_S_, respectively. Assume that the initial transformation parameters are R_O_ and T_O_;(2)Set the transformation parameters R_O_ and T_O_ and solve for the corresponding point P′ of point cloud P as follows:(4)$$P^{\prime }=R_{o}P_{i}+T_{o}$$

Find the point q_s_ in point cloud Q that has similar features to it, calculate the rigid body transformation matrices R and T for all nearest point pairs (p_s_,q_s_), and ensure that the objective function is minimized as follows:(5)$$f(R,T)=\frac{1}{n}\sum_{i=1}^{n}||q_{i}-(Rp_{i}+T){\left|\right|}^{2}$$

As shown in Equation (6), solve for the distance difference d between two adjacent point sets as follows:(6)$$d=|d_{n+1}-d_{n}|$$

In the equation, n is the number of iterations. When the number of iterations meets the threshold, the iteration stops; otherwise, steps (2) and (3) continue to be executed.

(3)Calculate the rigid body transformation matrices R and T to unify the coordinates of the point cloud data.

The combination of coarse and fine registration solves the problem of large-scale alignment and overall alignment accuracy. The RANSAC method effectively removes outliers and noise, while ICP further optimizes on this basis, improving registration accuracy. The initial transformation matrix provided by RANSAC allows ICP to converge quickly, reducing the number of iterations and improving computational efficiency. Ultimately, robust high-precision registration is achieved.

## 5. Results and Analysis

A total of 157 sets of collection experiments were conducted on 20 mutton sheep. During the experiments, the sheep briefly stayed in the limiting bars, and the system performed synchronous data collection and storage. Among these, 22 sets were discarded due to severe exposure caused by timing issues, resulting in significant data loss. Thus, 135 sets of valid depth maps were collected.

### 5.1. Point Cloud Collection and Reconstruction Results

The collected point clouds reflect the body condition of the sheep from various angles and do not contain significant interference noise points, making them suitable for 3D reconstruction. However, some point cloud data were incomplete or severely affected by noise due to improper filtering thresholds, and such point clouds were defined as failed collection data and were not used for the 3D reconstruction of the sheep body. The ideal data collected accounted for 85.98% of the total data, indicating that this method can effectively collect ideal point cloud data for 3D reconstruction. Figure 7 shows the reconstruction effects of the sheep body point cloud from three views after coarse and fine registration.

### 5.2. Reconstruction Accuracy

The point cloud coordinates collected by this system correspond to the actual distances. To verify the accuracy of the 3D point cloud reconstruction of the sheep body, the body length of the sheep was selected as a representative measure of the 3D point cloud reconstruction accuracy. Before the experiment, manual measurements were taken of the body length of all sheep, and the corresponding ear tag numbers were recorded. After the experiment, the distance tool in the Cloud Compare software was used to manually measure the body size data of the reconstructed sheep body.The version used is V2.13 Beta. The comparison of the two measurement results is shown in Table 2. Manual measurements were taken three times for each sheep and averaged, while software measurements were taken twice and averaged.

As shown in Table 2, the average error between the body length values of the 3D point cloud of the sheep reconstructed by this method, and the software measurements is 0.79% (taking body length as an example). This indicates that the point cloud reconstruction has a high degree of fidelity and accuracy, providing a reliable reference for the study of 3D point cloud reconstruction methods for sheep bodies and laying the foundation for non-contact measurement of body size and weight [28,29], which also used a similar method for validation and obtained good results, proving that this method can be used for data verification.

## 6. Discussion

The choice of point cloud filtering and registration strategies for the sheep body has a certain impact on the 3D point cloud reconstruction effect. The following issues are discussed in combination with the data collection process and the actual objective environment.

### 6.1. Impact of Outlier Filtering on Registration

In this study, outlier noise mainly comes from limiting devices, sensor errors, and exposure reflections. Statistical filtering was used to eliminate abnormal noise points around the sheep. In Section 4.1.1 (2), statistical filtering was used to set the threshold R based on statistical principles, and effective noise removal was achieved by adjusting the R value. By changing the R value to observe the test effect, when R > 0.6, the larger the value, the fewer points are removed, and the more redundant points remain; when R < 0.6, the smaller the value, the greater the intensity of point removal, resulting in defects in some areas. Through comprehensive testing, when R = 0.6, most noise points can be effectively eliminated while retaining the complete feature structure. Other scholars have different choices of the R value depending on the experimental animals. (Jiawei et al. chose R = 0.8 for processing cattle point clouds and obtained good results [30].)

### 6.2. Discussion on the Effect of a Conditional Voxel Filter Box

To solve the problems of redundant depth data and large data volume leading to slow computation, this study proposed a filtering algorithm combining conditional and voxel filtering. To evaluate the effect of conditional voxel filtering, the original point cloud of a single image with 14,285 data points was used as an example. The downsampling effects of using conditional filtering alone, voxel filtering alone, and conditional voxel filtering were compared. When the conditional filtering set area was the same and the voxel grid value was 0.5, the number of point clouds after downsampling was 11,173, 12,822, and 10,577, respectively. The downsampling effects are shown in Figure 5b–d.

Conditional filtering focuses on the attributes and feature selection of data, emphasizing the reduction in the limiting occlusion area in the experiment. Voxel filtering focuses more on the processing and optimization of spatial data, emphasizing the reduction in dense areas of the 3D spatial point cloud in the experiment. Based on the downsampling effects mentioned above, the combination of the two filters can achieve feature retention while further reducing the data volume, thus achieving better processing results. Researchers such as Zou et al. and Lyu et al. used different methods to achieve point cloud data downsampling design, each obtaining different results [31].

### 6.3. Discussion on the Effect of Target Rotation Normalization

During the image acquisition process, animals may stand in different positions within the confinement pen, causing their bodies to have varying tilt angles, as shown in Figure 8a,c. The angles of the sheep’s body significantly affect the reconstruction results and the measurement of the model’s body dimensions. Therefore, it is necessary to standardize and normalize the posture of the point cloud images to reduce the impact on the subsequent 3D reconstruction results. In this study, PCA (Principal Component Analysis) was used for point cloud rotation normalization. The covariance matrix was decomposed to obtain eigenvalues and eigenvectors. The eigenvectors were used as new coordinate axes, and the point cloud data were projected onto these axes, achieving rotation normalization. As shown in Figure 8b,c, this method effectively identifies the main direction of the point cloud data, resulting in better alignment in the new coordinate system.

To verify the effectiveness of the speed increase after the rotation normalization operation, a comparison was made between manual measurement speed, software measurement speed, and software measurement (normalized) speed, as shown in Table 3. Taking the measurement of six indicators for each sheep as the standard, it can be seen that there is a significant difference between manual measurement speed and software measurement speed. The average time for the manual measurement of one sheep is 7.25 min, while the average time for the software measurement of one sheep is 1.20 min, and the shortest measurement time after processing by the software group is 0.74 min. At the same time, the average measurement speed of each group also changes with the increase in the number of sheep. It is mainly manifested that the average measurement speed of the manual measurement group decreases with the increase in the number of sheep, while the average measurement speed of the untreated software group increases with the increase in the number of sheep. The analysis may be that the amount of data for calculation is too large, and the computer performance slows down. In the end, the software measurement (normalized) group performed the best, and its calculation speed was not affected after data processing.

### 6.4. Impact of Objective Factors on the Results

(1)The impact of wool length on the measurement error of reconstructed body size. There is a 0.79% error between the software body size measurement of the 3D reconstruction and the manual measurement, mainly due to the impact of wool length on the software measurement. During manual measurement, the wool needs to be compressed to minimize manual error; however, during machine measurement, the point cloud data from the three views are directly registered, resulting in the overall body length data being slightly larger than the manual measurement value. Therefore, further research on body size correction can be conducted in subsequent experiments.(2)The impact of limiting design on the reconstruction effect. Firstly, the design of the camera angles in the limiting bars is considered. The current design includes frontal, top, and side views, which can basically collect complete point cloud data of the sheep body. However, there may be angle deviations during the registration of some curved surfaces, significantly affecting the 3D reconstruction. When upgrading the limiting bars, adjusting the camera angles or adding more cameras can be considered to collect more comprehensive point cloud data for reconstruction. Secondly, the design of the outer edges of the limiting bars is considered. The current design uses a mesh steel structure, which generates a lot of noise during collection, making preprocessing and registration operations difficult. Therefore, different materials, such as using acrylic panels on one side, can be considered for future improvements to the limiting bars. This would provide unobstructed data collection while limiting and preparing for accurate 3D reconstruction of body size and weight estimation.

Overall, the following aspects can be improved in this study. Firstly, the choice and development of non-type filtering algorithms have a significant impact on the effectiveness of 3D reconstruction, so developing more ideal filtering algorithms is of great importance in practical production; secondly, for sheep, the main source of error in this study comes from the influence of wool length on measurement, and more research can be conducted on the body size correction of wool; and finally, a more rational design of the limiting fence can reduce image noise, which is of great significance for data processing and maintaining the integrity of the collected data.

## 7. Conclusions

This study utilizes a non-contact data acquisition system for the 3D point cloud reconstruction of sheep bodies. As the sheep pass through a specially designed trapezoidal limit bar, point cloud data from three angles are synchronously collected. Filtering algorithms are employed to denoise, repair, and register the point clouds, achieving the reconstruction of the 3D point cloud of the sheep and the measurement of body dimensions. For the first time, a conditional voxel filtering box is proposed for downsampling, which retains feature areas while effectively reducing the number of point clouds. A rotation normalization algorithm is used to correct the three views. Finally, a comparison is made between manual measurement speed, system measurement speed, and normalized system measurement speed. The results indicate that the normalized processing group has the most ideal time, with an average time of 0.74 min to measure six indicators for one sheep. Lastly, the accuracy of the 3D point cloud reconstruction of the sheep body is verified, with the reconstruction accuracy error of body length being 0.79%. This demonstrates that the non-contact body measurement method based on 3D point cloud reconstruction is feasible and can provide an important reference for the breeding of superior breeds.

## Figures and Tables

**Figure 1 animals-14-02457-f001:**
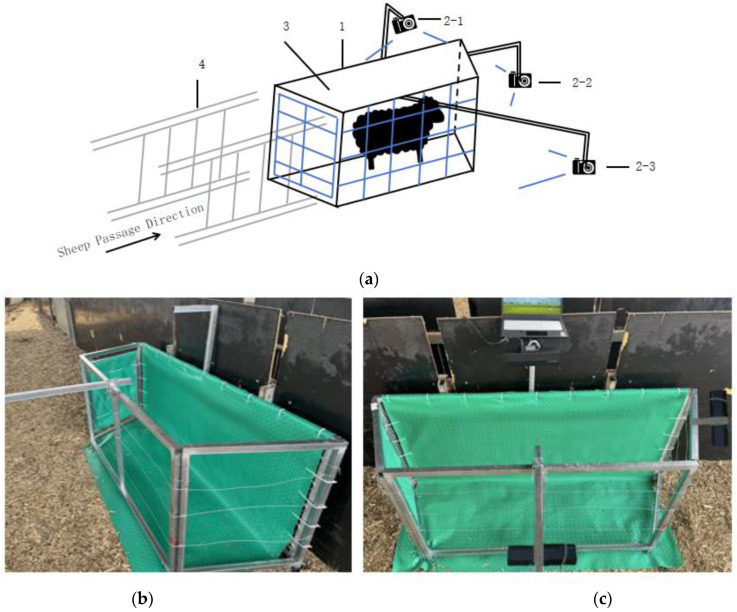
The sheep depth map data acquisition system, where (**a**) shows the system design diagram, with 1 being the trapezoidal limit bar, 2 being the depth camera, 3 being the background cloth, and 4 being the limit channel; (**b**,**c**) shows the physical diagram of the system.

**Figure 2 animals-14-02457-f002:**
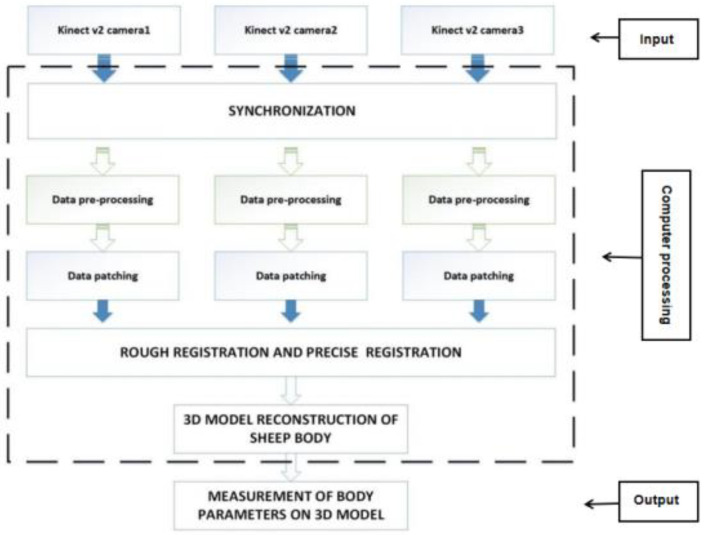
The flowchart of the non-contact measurement process of sheep body parameters using three Kinect v2 cameras.

**Figure 4 animals-14-02457-f004:**
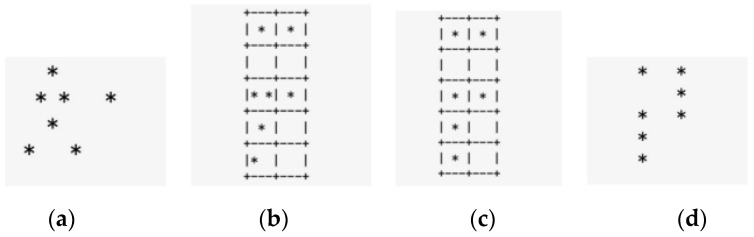
A schematic diagram of the point cloud voxel filtering process, where “*” represents a point cloud, (**a**) is the original point cloud, (**b**) is the grid division, (**c**) is the calculation of representative points, and (**d**) is the output of points.

**Figure 5 animals-14-02457-f005:**
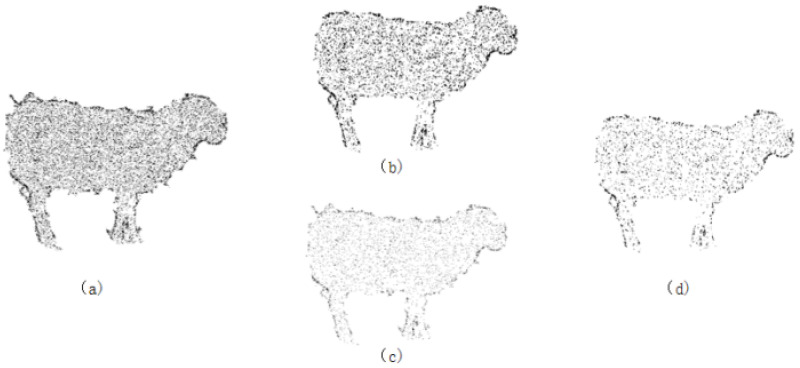
The effect before and after conditional voxel filtering downsampling, where (**a**) is the original point cloud before filtering, (**b**) is the effect of conditional filtering, (**c**) is the effect of voxel filtering, and (**d**) is the effect of conditional voxel filtering.

**Figure 6 animals-14-02457-f006:**
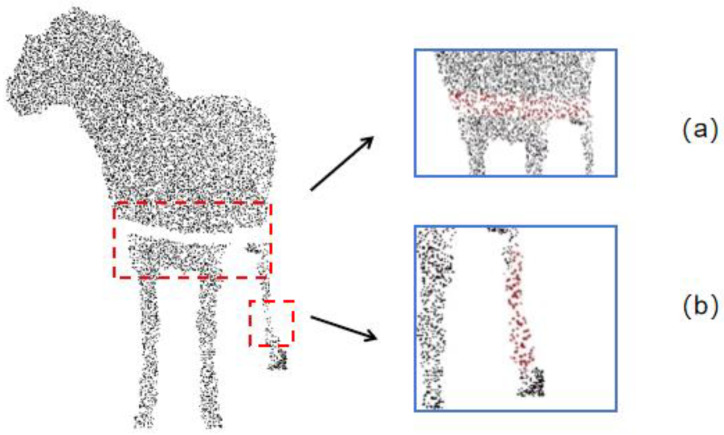
The effect of data patching in the front view, where the red dots in (**a**) represent the patched effect of the sheep’s chest area after being obstructed by the railing, and the red dots in (**b**) represent the patched effect of the sheep’s hind leg.

**Figure 7 animals-14-02457-f007:**
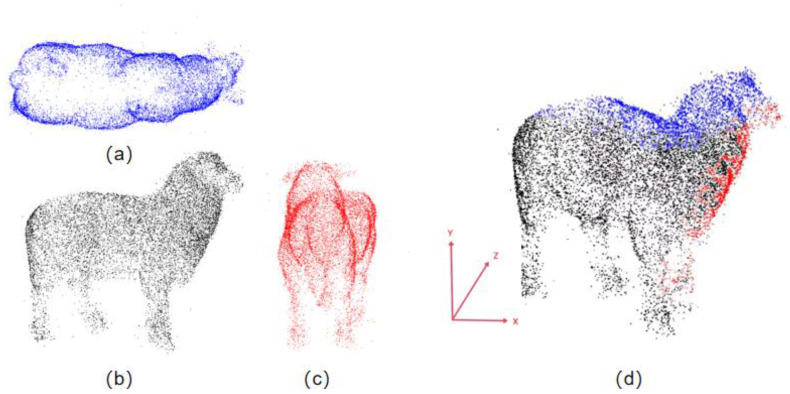
The effect of point cloud reconstruction from three views, where (**a**) is the top view of the sheep point cloud, (**b**) is the side view of the sheep point cloud, (**c**) is the front view of the sheep point cloud, and (**d**) is the reconstruction effect diagram of the point cloud from the three perspectives, with the points in three different colors each originating from the single perspectives of (**a**–**c**).

**Figure 8 animals-14-02457-f008:**
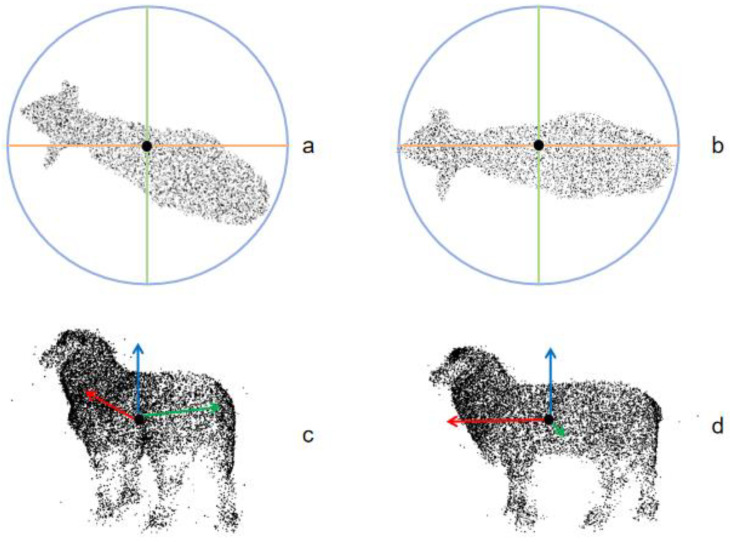
The effect of the sheep body rotation normalization algorithm is shown, with the black dot representing the axis center of the sheep body, and the surrounding rays indicating the coordinate directions. (**a**,**c**) are the top view and side view of the sheep body with an inclination angle, while (**b**,**d**) are the top view and side view of the sheep body after adjustment by the normalization algorithm.

**Table 1 animals-14-02457-t001:** Main parameters of the system design.

Parameter Category	Value
Passage Width × Height (T × H)/(mm)	600 × 900
Trapezoidal Restraining Pen Width × Height (T × H)/(mm)	600 × 900
Trapezoidal Restraining Pen Length a (L)/(mm)	1400
Trapezoidal Restraining Pen Length b (L)/(mm)	1900
Acute Angle of Trapezoidal Restraining Pen (A)/(°)	53°
Depth Camera Viewing Angle (H × V)/(°)	70° × 60°
Depth Camera Measurement Range (P)/(mm)	500–4500
Depth Camera Rod Length (L)/(mm)	900

**Table 2 animals-14-02457-t002:** The verification results of point cloud measurement accuracy, with a measurement error of 0.79% when comparing the software measurement results with traditional measurement.

Serial Number	Tape Measurement (cm)	Point Cloud Measurement (cm)	Error/%	Serial Number	Tape Measurement (cm)	Point Cloud Measurement (cm)	Error/%
1	67.1	66.0	1.7	11	62.4	62.5	0.2
2	64.8	64.0	1.3	12	64.7	64.0	1.1
3	67.6	67.5	0.1	13	68.1	67.5	0.9
4	70.4	71.0	0.8	14	60.7	60.5	0.3
5	62.6	62.0	1.0	15	66.5	66.0	0.8
6	70.7	70.5	0.3	16	64.9	64.0	1.4
7	74.1	74.0	0.1	17	65.2	64.5	1.1
8	60.6	60.0	1.0	18	63.0	62.7	0.5
9	62.8	62.0	1.3	19	64.3	64.0	0.5
10	67.3	67.0	0.4	20	63.7	63.0	1.1

**Table 3 animals-14-02457-t003:** The comparison of manual measurement speed, software measurement speed, and software measurement speed (after normalization treatment) following the rotation normalization operation.

Sheep Population	Manual (min)	Average Measurement Speed (min)	Software (min)	Average Measurement Speed (min)	Software (Post-Processing) (min)	Average Measurement Speed (min)
1 sheep	7.25	7.25	1.20	1.20	0.74	0.74
20 sheep	138.00	6.90	24.70	1.24	14.60	0.73
50 sheep	317.00	6.34	67.70	1.35	36.50	0.73

## Data Availability

None of the data were deposited in an official repository. The data that support the study findings are available from the authors upon request.

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
