# Peer review of "A Study on the 3D Reconstruction Strategy of a Sheep Body Based on a Kinect v2 Depth Camera Array"

_animals, 2024, doi:10.3390/ani14172457_

Round 1

Reviewer 1 Report

Comments and Suggestions for Authors

Row 9: Word “me asurement” should be written as “measurement”

Row 15: Word “da1ta” should be written as “data1

Rows from 72 to 78: The aim of paper should be explained more precisely.

Row from 448 to 461: In conclusion, it would be wise to mention the next steps for research or the questions that urgently need to be answered.

Author Response

Thank you for your suggestions for revision. Below is the updated situation:

Review1:Row 9: Word “me asurement” should be written as “measurement”

Reply1:Spelling error has been corrected to“measurement”

Review2:Row 15: Word “da1ta” should be written as “data1”

Reply2:There was a formatting issue, which has been changed to“data1”

Review3:Rows from 72 to 78: The aim of paper should be explained more precisely.

Reply3:The suggestions were accurately described for this part:“Traditional animal body measurement methods primarily rely on manual data collection, which is not only costly and inefficient but also prone to significant data errors and can cause stress to the animals. To address these issues, this study proposes a 3D point cloud reconstruction method for sheep bodies, achieving synchronous acquisition and reconstruction of sheep body point cloud data. The reconstruction accuracy meets the relevant body measurement standards, providing a theoretical basis for the measurement of sheep body phenotype data.”

Review4:Row from 448 to 461: In conclusion, it would be wise to mention the next steps for research or the questions that urgently need to be answered.

Reply4:For the outlook and next steps in research, the following new content has been added:“Overall, the following aspects can be improved in this study. Firstly, the choice and development of non-type filtering algorithms have a significant impact on the effectiveness of 3D reconstruction, so developing more ideal filtering algorithms is of great importance in practical production; secondly, for sheep, the main source of error in the study comes from the influence of wool length on measurement, and more research can be conducted on the body size correction of wool; finally, a more rational design of the limiting fence can reduce image noise, which is of great significance for data processing and maintaining the integrity of the collected data.”

Thanks

Reviewer 2 Report

Comments and Suggestions for Authors

The authors provide a non-contact method for measuring frontobody length with point cloud reconstruction of sheep body, which is critical to advancing non-contact measurement and welfare of animals. There are some minor revisions needs to revised.

- In order to solve the problem of slow depth data calculation, the author can give more comparative indicators, such as algorithm calculation speed.

- Although the author only calculates the body length, if the body height is other body size, does the author have a preliminary result to prove the advantages and disadvantages of the algorithm for other body size measurements?

- It is better to give a diagram of software measurement and manual measurement.

- The horizontal coordinate in Figure 9 seems wrong, please correct it.

- You can also give this reference to strengthen body measurement in other animals in the conclusion also. MCP: Multi-Chicken Pose Estimation Based on Transfer Learning DOI: 10.3390/ani14121774

Author Response

Review1:In order to solve the problem of slow depth data calculation, the author can give more comparative indicators, such as algorithm calculation speed.

Reply1:Thank you for the constructive suggestions. This point will be presented in the form of a new table in the text, showing the measurement speeds of different methods.(The following modified content has been changed to red in the article)

Review2: Although the author only calculates the body length, if the body height is other body size, does the author have a preliminary result to prove the advantages and disadvantages of the algorithm for other body size measurements?

Reply2:This study conducted multiple indicator tests on sheep bodies, including body length, height, chest depth, chest width, and buttock height. Due to the limitation of article length, only 'body length' was selected to display the measured values.

Review3:It is better to give a diagram of software measurement and manual measurement.

Reply3:This is a good suggestion to compare the measurement speeds under different forms, and this part of the content has already been added to the text.

Review4:The horizontal coordinate in Figure 9 seems wrong, please correct it.

Reply4:Figure 9 was not persuasive enough and has been replaced by Table 3.

Review5:You can also give this reference to strengthen body measurement in other animals in the conclusion also. MCP: Multi-Chicken Pose Estimation Based on Transfer Learning DOI: 10.3390/ani14121774

Reply5:This article differs from the direction of my paper, but I can learn from its research and analysis methods, which I have already cited in my work. Based on suggestions, I have made some changes to my conclusion.

Thanks

Reviewer 3 Report

Comments and Suggestions for Authors

Simple Summary is missing.

You don’t need to write a chapter titled “Implication.” The first chapter is the introduction.

The main question addressed by the research is how to develop a non-contact 3D point cloud reconstruction method for sheep bodies using a Kinect v2 depth camera array. The study aims to create a system that can accurately and efficiently collect and process sheep body data for breeding and fattening purposes without causing animal stress during manual measurement. The study also explores methods to mitigate environmental noise and other factors that could affect the accuracy of data collection in practical production settings.

 The original and relevant parts of the study include the development of a non-contact 3D reconstruction method specifically for sheep bodies using a Kinect v2 depth camera array. This innovative method combines algorithms such as passthrough filtering, statistical filtering, and Random Sample Consensus (RANSAC) to handle noise in point cloud data. The study also introduces a conditional-voxel filtering box for downsampling the point cloud data, which helps retain key features while reducing data volume. Using coarse and fine registration techniques, including RANSAC and Iterative Closest Point (ICP) algorithms, further enhances the accuracy and robustness of the 3D reconstruction.

The specific gap addressed by this paper is the need for a practical, accurate, and stress-free method for measuring sheep body dimensions in real-world production environments. Traditional manual measurement methods are prone to errors, are time-consuming, and cause animal stress, affecting their welfare. This study provides a non-invasive solution that could improve the accuracy and efficiency of livestock management, particularly in breeding and fattening processes.

Review writing, missing spaces (especially after the reference citations), etc...

The introduction is well-written, with some literature related to methods of measuring sheep body dimensions.

 The methodology involves multiple steps and techniques, which may make it complex to implement in less controlled environments or by users without significant technical expertise.

Overall, the methodology is thorough and well-suited for the study’s objectives, addressing the specific challenges of non-invasive, accurate measurement of livestock in real-world conditions.

The discussion may compare the results with other references.

The conclusions are robust, forward-thinking, and well-connected to the research findings. They successfully communicate the study's contributions to the scientific community and practical applications in livestock management.

The references are appropriate; they cover many publications within the studied area.

The tables are ok; Some Figures must be replaced, like Figure 1c

Author Response

Review1:Simple Summary is missing.

Reply1:This part has already been supplemented.

Review2:You don’t need to write a chapter titled “Implication.” The first chapter is the introduction.

Reply2:The modifications have been made according to the suggested format.

Review3:Review writing, missing spaces (especially after the reference citations), etc...

Reply3:According to the suggestions, the format has been modified

Review4:The methodology involves multiple steps and techniques, which may make it complex to implement in less controlled environments or by users without significant technical expertise.

Reply4:This study's next step is to simplify the system and algorithms, reduce the impact of complex environments on related work, and further enhance the practicality of the research.

Review5:The discussion may compare the results with other references.

Reply5:Following the suggestions, to improve the discussion section, new content has been added to this section regarding the comparison of rates between different measurement methods, and similar references have been cited for reference.

Review6:Some Figures must be replaced, like Figure 1c

Reply6:Figure 1C has been replaced.

Thanks